# Adipocyte-Specific ACKR3 Regulates Lipid Levels in Adipose Tissue

**DOI:** 10.3390/biomedicines9040394

**Published:** 2021-04-06

**Authors:** Selin Gencer, Yvonne Döring, Yvonne Jansen, Soyolmaa Bayasgalan, Olga Schengel, Madeleine Müller, Linsey J. F. Peters, Christian Weber, Emiel P. C. van der Vorst

**Affiliations:** 1Institute for Cardiovascular Prevention (IPEK), Ludwig-Maximilians-Universität München, 80337 Munich, Germany; Selin.Gencer@med.uni-muenchen.de (S.G.); yvonne.doering@med.unibe.ch (Y.D.); Yvonne.Jansen@med.uni-muenchen.de (Y.J.); Soyolmaa.Bayasgalan@med.uni-muenchen.de (S.B.); Olga.Schengel@med.uni-muenchen.de (O.S.); madeleine.mueller@tum.de (M.M.); lipeters@ukaachen.de (L.J.F.P.); Christian.Weber@med.uni-muenchen.de (C.W.); 2DZHK (German Center for Cardiovascular Research), Partner Site Munich Heart Alliance, 80337 Munich, Germany; 3Swiss Cardiovascular Center, Department of Angiology, Inselspital, Bern University Hospital, University of Bern, 3010 Bern, Switzerland; 4Institute of Molecular Immunology, TUM School of Medicine, Technical University of Munich, 80333 Munich, Germany; 5Interdisciplinary Center for Clinical Research (IZKF), Institute for Molecular Cardiovascular Research (IMCAR), RWTH Aachen University, 52056 Aachen, Germany; 6Department of Pathology, Cardiovascular Research Institute Maastricht (CARIM), Maastricht University, 6200 Maastricht, The Netherlands; 7Department of Biochemistry, Cardiovascular Research Institute Maastricht (CARIM), Maastricht University Medical Centre, 6229 Maastricht, The Netherlands; 8Munich Cluster for Systems Neurology (SyNergy), 81377 Munich, Germany

**Keywords:** ACKR3, adipocyte, adipose tissue, lipid, metabolism, hyperlipidemia, PPAR-γ, ANGPTL4

## Abstract

Dysfunctional adipose tissue (AT) may contribute to the pathology of several metabolic diseases through altered lipid metabolism, insulin resistance, and inflammation. Atypical chemokine receptor 3 (ACKR3) expression was shown to increase in AT during obesity, and its ubiquitous elimination caused hyperlipidemia in mice. Although these findings point towards a role of ACKR3 in the regulation of lipid levels, the role of adipocyte-specific ACKR3 has not yet been studied exclusively in this context. In this study, we established adipocyte- and hepatocyte-specific knockouts of *Ackr3* in ApoE-deficient mice in order to determine its impact on lipid levels under hyperlipidemic conditions. We show for the first time that adipocyte-specific deletion of *Ackr3* results in reduced AT triglyceride and cholesterol content in ApoE-deficient mice, which coincides with increased *peroxisome proliferator-activated receptor-γ (PPAR-γ)* and increased *Angptl4* expression. The role of adipocyte ACKR3 in lipid handling seems to be tissue-specific as hepatocyte ACKR3 deficiency did not demonstrate comparable effects. In summary, adipocyte-specific ACKR3 seems to regulate AT lipid levels in hyperlipidemic *Apoe*^−/−^ mice, which may therefore be a significant determinant of AT health. Further studies are needed to explore the potential systemic or metabolic effects that adipocyte ACKR3 might have in associated disease models.

## 1. Introduction

Adipose tissue (AT) was traditionally regarded as a dormant fat tissue; however, it is now well understood that AT is a highly active metabolic organ exerting numerous vital functions in the body. In addition to its well-known lipid storage function, glucose homeostasis, hormone secretion, energy homeostasis and thermogenesis are examples of further tasks accomplished by the AT [1,2]. While lipid storage is a natural function of AT, excess lipid accumulation within the tissue, for example, as a result of abnormally high lipid levels in the blood (hyperlipidemia), may burden the adipocytes. Subsequently, this burden may compromise healthy AT functions and lead to detrimental complications, such as inflammation and metabolic stress [3]. Accumulating evidence reveals a significant role of dysfunctional AT in the pathology of several diseases, such as obesity and insulin resistance, highlighting AT as a significant contributor to the metabolic syndrome, as well as cardiovascular diseases (CVDs) [4,5,6]. In addition to the AT, liver is another major metabolic organ regulating the lipid metabolism in the body and over-accumulation of lipids within the liver likewise leads to highly detrimental metabolic complications, such as the non-alcoholic fatty liver disease (NAFLD). NAFLD is highly associated with insulin resistance and obesity and patients suffering from these conditions are at higher risk for CVDs [7].

While some medical options are available to treat patients suffering from hyperlipidemia (e.g., statins) in order to prevent subsequent complications as mentioned above, the search for more effective therapeutic options is still ongoing. The chemokine system plays a fundamental role in health and disease, and they have been shown to contribute markedly to the pathology of chronic inflammatory diseases, such as atherosclerosis [8], as well as autoimmune diseases and cancer [9]. Furthermore, the chemokine system is also recognized to play important roles in the development of obesity and insulin resistance [10]. Identifying specific chemokine-receptor axes regulating pathological processes in the root of these diseases, such as excessive lipid accumulation in tissues, may allow us to manipulate these axes for therapeutic purposes. A potential candidate in the research of metabolic diseases is the atypical chemokine receptor 3 (ACKR3).

ACKR3, previously known as CXCR7, is the alternate receptor of the inflammatory chemokine CXCL12, and several studies have unveiled the importance of this chemokine receptor axis in cancer as well as cardiometabolic diseases [11]. Additionally, ACKR3 was shown to be upregulated in AT during obesity in Western diet (WD)-fed mice, suggesting that adipocyte ACKR3 may be involved in the pathology of obesity [12]. Moreover, systemic genetic ablation of ACKR3 was shown to cause hyperlipidemia in *Apolipoprotein E deficient* (*Apoe*^−/−^) mice, whereas its activation through CCX771 injections in mice decreased serum cholesterol and triglyceride levels [13]. This effect was attributed to increased very-low-density lipoprotein (VLDL) uptake by the AT, suggesting that ACKR3 controls systemic lipid levels through AT lipid uptake [13].

These findings suggest that ACKR3 may play an important role in the regulation of AT lipid levels, which may therefore affect AT health. However, the research in this area is scarce and so far only systemic approaches have been used to study this concept. Furthermore, liver-specific effects of ACKR3 are not known. In order to fill this gap and to pinpoint cell-specific effects of ACKR3, we developed adipocyte and hepatocyte-specific ACKR3-deficient mice on an *Apoe*^−/−^ background, which are prone to develop hyperlipidemia on WD, to determine whether ACKR3 can alter tissue lipid levels.

## 2. Materials and Methods

### 2.1. Mice

Adipoq-cre mice expressing the Cre recombinase under the control of the adipocyte-specific protein adiponectin were purchased from Jackson Laboratories (Bar Harbor, ME, USA; Stock# 025124) and crossed with Apolipoprotein E-deficient (*Apoe*^−/−^) mice to generate *AdipoqCreApoe*^−/−^ mice. These mice were then crossed with ACKR3 floxed (*Ackr3^fl/fl^*) *Apoe*^−/−^ mice to generate *AdipoqCreAckr3^fl/fl^Apoe*^−/−^ mice for an adipose tissue-specific knockout of ACKR3. The knockout was induced with daily tamoxifen injections (Sigma; 1.5 mg per 20 g body weight, dissolved in corn oil) for 5 consecutive days (Appendix A), followed by a 4-week WD containing 21% fat and 0.15% to 0.2% cholesterol (Sniff diets). Liver-specific knockout of ACKR3 was achieved by injecting 1 × 10^11^ AAVs (AAV8 expressing iCre driven by a liver ALB (1.9) promoter, Vector Biolabs) into *Ackr3^fl/fl^Apoe*^−/−^ mice (Appendix A). Mice were fed with a 4-week WD as described above. All mice were on a C57BL/6J background. All animals were bred and housed in the local animal facility under specific pathogen free (SPF) conditions. Prior to the start of the WD, all mice were fed a normal chow diet. All animal experiments were approved by the local ethical committee (Regierung von Oberbayern, Sachgebiet 54, Germany; ROB-55.2-2532.Vet_01-18-96). All methods were carried out in accordance with relevant guidelines and regulations.

### 2.2. Tissue Homogenization

After sacrifice, mice were fully perfused with PBS before organ collection in order to eliminate blood from the tissues. Adipose tissue and liver samples were weighed and washed with ice cold PBS (Gibco). Samples were homogenized in a bead-based tissue lyser using 500 µL cell lysis buffer (Cell Signaling Technology, Danvers, MA, USA) containing protease and phosphatase inhibitors (Roche, Basel, Switzerland) per 250 mg of tissue. Samples were then centrifuged at 10,000× *g* for 10 min at 4 °C. Total protein levels in the homogenized tissue were measured via a commercially available BCA assay kit (Pierce™ BCA™ Protein-Assay from ThermoFisher Scientific, Waltham, MA, USA) according to the manufacturer’s protocol.

### 2.3. Lipid Measurement

Cholesterol and triglyceride levels were quantified in EDTA-plasma and homogenized tissue using enzymatic assays (c.f.a.s. cobas, Roche Diagnostics) according to the manufacturer’s protocol.

### 2.4. LPL Activity

Adipose tissue homogenate lipoprotein lipase activity was quantified by lipoprotein lipase assay kit (Fluorometric; ab204721) purchased from Abcam according to the manufacturer’s protocol. LPL activity was then normalized to total protein levels of the tissue homogenates.

### 2.5. HPLC

Plasma samples were subjected to fast-performance liquid chromatography (gel filtration on Superose 6 column (GE Healthcare)). Different lipoprotein fractions (very low densitiy lipoprotein (vLDL), low-density lipoprotein (LDL), and high-density lipoprotein (HDL) were separated and evaluated based on their retention (flow-through) times as follows: vLDL between 40 and 50 min, LDL between 50 and 70 min, and HDL between 70 and 90 min. Total cholesterol levels of the plasma samples were quantified as described above.

### 2.6. RNA Isolation

Total RNA isolation was performed by commercially available RNA isolation kit from Zymoresearch according to the manufacturer’s protocol. Direct-zol miniprep kit was used for RNA isolation from tissues such as the liver and the adipose tissue, whereas the Direct-zol microprep kit was used for RNA isolation from cultured cells. The quality (A_260_/A_280_) and the quantity (ng/µL) of the RNA were measured by NanoPhotometer N60/N50 (Implen). A ratio of ~2 for A_260_/A_280_ was accepted as good quality RNA.

### 2.7. cDNA Synthesis

RNA samples were diluted to the same concentration, and the cDNA synthesis was performed via the commercially available iScript cDNA synthesis kit from Bio-Rad according to the manufacturer’s protocol.

### 2.8. Droplet Digital PCR

PCR was performed on QX200 Droplet Digital PCR (ddPCR™) system from Bio-Rad. Twenty µL reaction mixes were prepared using 10 μL of 2× ddPCR SuperMix for probes (No dUTP) (Bio-Rad, Hercules, CA, USA), 1 μL of 20× FAM labeled primer/probe for the target gene, 1 μL of 20× VIC labeled primer/probe for the housekeeping gene (*Gapdh* or *18S*), RNase-/DNase-free water, and cDNA sample. Taqman primers were purchased from Thermofisher Scientific. Droplet generation was performed in the QX200 droplet generator (Bio-Rad) by adding 20 μL reaction mix and 70 μL droplet generation oil for probes (Bio-Rad) onto DG-8 cartridges covered with gaskets (Bio-Rad). Forty-two μL of the droplet solution (containing up to 20,000 droplets) was transferred to the appropriate PCR plate (Bio-Rad), which was then sealed with a piercable foil using the PCR plate sealer (Bio-Rad). Cycling was performed in the ddPCR cycler with the following conditions: 10 min at 95 °C (enzyme activation), 30 s at 94 °C (denaturation), and 1 min at 60 °C (annealing/extension) for 40 cycles, and 10 min at 98 °C (enzyme deactivation). The PCR plate was then proceeded to the droplet reader (Bio-Rad). Analysis was performed on QuantaSoft software (Bio-Rad).

### 2.9. Western Blot

Equal amounts of protein samples obtained from tissue homogenates were mixed with Novex™ NuPAGE™ LDS sample buffer (4×) and boiled at 95 °C for 5 min. Lysates were then loaded onto 12% Mini-PROTEAN^®^ TGX™ Precast Protein Gels (Bio-Rad) along with Precision Plus Protein™ WesternC™ blotting standard (Bio-Rad). Western blot was performed with Mini Trans-Blot^®^ Cell and Criterion™ Blotter according to the manufacturer’s protocol (Bio-Rad). After blocking the membrane (5% bovine serum albumin (BSA)), primary antibody incubation against rabbit polyclonal phospho-PPAR-γ S112 (from Mybiosource, San Diego, CA, USA) was carried out overnight at 4 °C. The HRP-labelled anti-rabbit secondary antibody (from Abcam, Cambridge, UK) incubation was carried out for 1 h at room temperature. The membrane was developed with SuperSignal™ West Pico PLUS Chemiluminescent Substrate (from Thermofisher). After the signal for phospho- PPAR-γ was captured, the membrane was stripped for 30 min with Restore™ Western Blot Stripping Buffer (from Thermofisher) and blocked for 1 h with 5% BSA solution. The membrane was then incubated with recombinant anti-PPAR-γ antibody (from Abcam) for 1 h at room temperature in order to detect total PPAR-γ levels, and this was followed by again 1-h incubation of secondary antibody. The signal was captured as explained above. Analysis of phospho and total protein signals was performed on ImageJ, and the phospho protein signal was normalized by dividing it to total protein signal.

### 2.10. Flow Cytometry

Whole blood was collected from mice in EDTA-buffer tubes, and red blood cell lysis was performed prior to stainings. Hematopoietic cell populations in blood samples were stained as follows: anti-CD45, anti-CD115, anti-Gr1, anti-CD11b, anti-B220, anti-CD3, anti-CD4, and anti-CD8. Cell suspensions were analyzed with FACS Canto II together with FACSDiva software (BD Biosciences). Cell populations were gated and analyzed as follows on FlowJo Software: leukocytes (CD45+), neutrophils (CD45 + CD115-Gr1high), monocytes (CD45 + CD11b + CD115+), T cells (CD45 + CD3+), and B cells (CD45 + B220+).

### 2.11. ELISA Assays

Plasma levels of IL-6 and TNF-α were quantified from EDTA plasma by ELISA using a commercially available kit (Uncoated ELISA kit, ThermoFisher Scientific) according to the manufacturer’s protocol. Plasma levels of CXCL12 were quantified in the same fashion using a commercially available kit (Quantikine ELISA kit, R&D Systems, San Diego, CA, USA).

### 2.12. Statistics

All data are expressed as mean ±SEM. Statistical analyses were performed using GraphPad Prism version 7.0 or higher (GraphPad Software, Inc., San Diego, CA, USA). Outliers were identified with ROUT = 1 and normality of the data was tested via the D’Agostino-Pearson omnibus normality test. Statistical significance was tested via unpaired Student’s *t*-test with Welch correction for normally distributed data and Mann-Whitney U test for non-normally distributed data. A result of < 0.05 for *p*-value was considered statistically significant. * *p* < 0.05, ** *p* < 0.01, *** *p* < 0.0013. 

## 3. Results

### 3.1. Adipocyte-Specific Ackr3 Deficiency Decreases AT Lipid Content in Western Diet Fed Mice

*AdipoqCre^+^Ackr3^fl/fl^Apoe*^−/−^ and *AdipoqCre^−^Ackr3^fl/fl^Apoe*^−/−^ (control) mice were fed a WD for 4 weeks (Figure 1a), and visceral white adipose tissue (wAT) lipid levels were quantified. *AdipoqCre^+^Ackr3^fl/fl^Apoe*^−/−^ mice had significantly lower levels of total triglyceride as well as total cholesterol in wAT compared to control mice (Figure 1b,c). However, plasma triglyceride and cholesterol levels did not differ between the two groups (Figure 1d,e). Lipid content in the plasma samples was further analyzed by quantifying the lipoprotein fractions VLDL, LDL, and HDL, and no differences were observed between the two genotypes (Figure 1f–i). Likewise, there were no differences in hepatic triglyceride and cholesterol levels (Figure 1j,k), indicating that the lack of adipocyte *Ackr3* only has local effects on the lipid metabolism. 

No differences could be observed in general characteristics of the mice, such as the body weight, visceral wAT weight, the quantity of circulating immune cells, and inflammatory plasma cytokine levels including ACKR3 ligand CXCL12 (Table 1).

General characteristics in control and knockout mice. Data are presented as mean ± SEM. Immune cell subsets were analyzed by flow cytometry (*n* = 14–15). Mice were weighed before the sacrifice (*n* = 14–15). Plasma cytokines were measured via ELISA (*n* = 7–18). Visceral wAT samples were weighed after the sacrifice (*n* = 6–8).

### 3.2. Expression of Lipid Receptors in wAT Samples

In order to examine how ACKR3 affects lipid levels in the AT, RNA was isolated from the wAT of *AdipoqCre^+^Ackr3^fl/fl^Apoe*^−/−^ and *AdipoqCre^−^Ackr3^fl/fl^Apoe*^−/−^ mice, and the gene expression of well-known receptors that are involved in the lipid metabolism was examined via ddPCR. No significant differences were detected in the expression of *Abcg1, Abca1, Ldlr,* and *Sort1* between the two groups (Figure 2a–d), whereas *Vldlr* expression was significantly increased in the knockout mice, reflecting a potential compensatory mechanism (Figure 2e).

### 3.3. Lack of ACKR3 in AT Increases Angptl4 and PPAR-γ Expression

In order to further assess the impact of *Ackr3* knockout on potential mechanisms regulating lipid handling in wAT samples, the expression of *Angtpl4* and *Ppar-γ* was examined via ddPCR. Both genes were found to be expressed significantly higher in the wAT of *AdipoqCre^+^Ackr3^fl/fl^Apoe*^−/−^ mice compared to control (Figure 3a,b). Consistent with the gene expression data, PPAR-γ phosphorylation was increased in the wAT of *AdipoqCre^+^Ackr3^fl/fl^Apoe*^−/−^ mice compared to control (Figure 3c,d). ANGPTL4 and PPAR-γ are key players in triglyceride metabolism as they negatively regulate the activity of lipoprotein lipase (LPL), thereby limiting the lipid uptake into the tissue [14,15]. Thus, we assumed that the elimination of adipocyte-ACKR3 may have limited the lipid uptake by interfering with LPL activity in AT through the activation of PPAR-γ and ANGPTL4. In line with this theory, the enzymatic activity of LPL in wAT samples correlated significantly with the wAT triglyceride levels (Figure 3e), suggesting that the ACKR3 dependent changes in wAT lipids are conceivably mediated by LPL.

### 3.4. Hepatic Ackr3 Deficiency Does Not Impact Local or Systemic Lipid Levels

To determine whether the lipid modulating effects of ACKR3 are specific to adipocytes, we also evaluated the effects of hepatocyte ACKR3 on local and systemic lipid levels as the liver is one of the main organs involved in the lipid metabolism. In order to study the impact of hepatic ACKR3 on tissue and systemic lipid levels, *AlbuminCre^+^Ackr3^fl/fl^Apoe*^−/−^ (Cre insertion via AAV8 injection) and *AlbuminCre^−^Ackr3^fl/fl^Apoe*^−/−^ mice were fed a WD for 4 weeks (Figure 4a) and lipid levels were quantified in the liver (Figure 4b,c), wAT (Figure 4d,e) and plasma (Figure 4f,g). Hepatic *Ackr3* deficiency did not lead to any local or systemic differences of total cholesterol and triglyceride levels compared to control mice, indicating that the observed role of ACKR3 on lipid uptake is adipocyte-specific.

## 4. Discussion

In this study, we generated mice that specifically lacked ACKR3 in adipocytes (*AdipoqCre^+^ACKR3^fl/fl^Apoe*^−/−^) in order to investigate its impact on AT lipid levels under hyperlipidemic conditions. According to our findings, adipocyte-specific ACKR3 is indeed involved in the regulation of AT lipid levels. Genetic deficiency of this receptor in adipocytes decreased lipid levels and increased the expression of *Angptl4* and *PPAR-γ*, as well as the phosphorylation of PPAR-γ protein in the AT of hyperlipidemic mice. Accordingly, AT lipid levels correlated positively with AT LPL activity. PPAR-γ increases the expression of ANGPTL4 [15,16], and ANGPTL4 is known to inhibit LPL activity [14]. Collectively, these results suggest that adipocyte ACKR3 regulates AT lipid accumulation by inhibiting the ANGPTL4 based restriction of LPL activity through PPAR-γ, which is also in line with the findings of the study by Li et al. [13]. Li et al. reported enhanced cholesterol uptake by the AT in hyperlipidemic mice that were systemically treated with CCX771 (described as an ACKR3 agonist), along with a decrease in serum lipid levels [13]. In addition, the study showed that CCX771 treatment decreased the ANGPTL4 protein levels as well as the expression of its positive regulator *PPAR-γ* in AT. The authors also concluded that ACKR3 activation increased the lipase activity in AT by decreasing ANGPTL4. However, these findings were based on a systemic ACKR3 agonist treatment and did not disclose whether or not adipocyte-specific ACKR3 was responsible for these effects. Besides, there are controversies in the literature regarding the effects of CCX771 on ACKR3; some studies report CCX771 based inhibition/antagonism of ACKR3 [17,18,19,20], whereas some studies describe it as an ACKR3 agonist [13,21]. Therefore, studying this role of ACKR3 in a cell-specific genetic knockout model provided novel insights and a rather precise verification to these findings.

Further on the subject, the impact of ACKR3 knockout on PPAR-γ activation is an outstanding observation. Previously, a PPAR-γ stimulating drug used in type 2 diabetes mellitus treatment (pioglitazone) was shown to inhibit ACKR3 expression in differentiated macrophages which resulted in suppressed chemotaxis [22]. According to our findings and that of Li et al., not only does PPAR-γ appear to negatively regulate ACKR3 but the same seems to be true the other way around, suggesting that there may be a potential feedback mechanism in action. Considering that the ACKR3 ligand CXCL12 is an inflammatory chemokine, it is not surprising that the anti-inflammatory PPAR-γ is upregulated in the absence of ACKR3. Furthermore, PPAR-γ is involved in the inhibition of the mitogen activated protein kinase (MAPK) pathway [23] and ACKR3 has been shown to signal through the MAPK pathway [24,25]. Nevertheless, the exact mechanism via which ACKR3 and PPAR-γ affect each other remains to be determined and will be an interesting focus point in future studies.

In contrast to our expectations and to the effects that were observed by systemic ACKR3 targeting [13], downregulation of AT lipid levels did not result in increased systemic lipid levels in our adipocyte-specific knockout model. One potential explanation for this discrepancy is the notion that LPL is not only expressed in AT but also abundantly in muscle tissues (MT), and, interestingly, LPL activity has been shown to be inversely regulated between AT and MT based on starvation and refeeding studies [26]. Therefore, the LPL activity in MT may be enhanced in contrast to AT in our model, which may not be the case in a model of systemic ACKR3 targeting used by Li et al. Another possibility is that ACKR3 expressed by another cell type may be contributing to the systemic changes in lipid levels, although we already ruled out involvement of hepatic ACKR3 in this. Furthermore, it is important to note that our mice were not fasted before the sacrifice. ANGPTL4-mediated differences in plasma lipid levels may depend on fed/fasted state as a study revealed that fasted but not fed *Angptl4*^−/−^ mice showed decreased triglyceride levels in the plasma [27]. The potential impact of ACKR3-driven decrease in AT lipid content on plasma lipid levels may have therefore been overshadowed in our study, which should be further elucidated in the future.

AT is a central metabolic organ that regulates significant processes in the body, such as lipid storage and energy balance. Disruption of healthy AT functions due to excessive lipid accumulation has been associated with AT inflammation and metabolic diseases, such as obesity [28,29]. The finding that ACKR3 knockout effectively reduces lipid uptake in the AT under hyperlipidemia reflects a beneficial physiological advantage for the AT as this process intervenes with excess lipid accumulation in the tissue, which could otherwise lead to a dysfunctional AT. Thus, ACKR3 may be detrimental for AT physiology under hyperlipidemic conditions.

Lastly, it is important to note that our findings are based on a hyperlipidemic mouse model (*Apoe*^−/−^ background), which may be a limitation of this study. Although hyperlipidemia is present in many patients and thus reflects a highly relevant model, the effects of adipocyte ACKR3 under normolipidemic conditions remain to be determined.

## 5. Conclusions

In this study, we report that adipocyte-specific ACKR3 seems to be involved in the regulation of AT lipid levels under hyperlipidemic conditions in *Apoe*^−/−^ mice. Furthermore, our data indicate that this role of adipocyte-ACKR3 is most likely mediated by PPAR-γ and ANGPTL4-regulated LPL activity in AT. These findings highlight the potential of ACKR3 in lipid metabolism and suggest that ACKR3 may be a possible contributor to metabolic diseases, such as insulin resistance, obesity, and atherosclerosis, which remains to be validated in future studies.

## Figures and Tables

**Figure 1 biomedicines-09-00394-f001:**
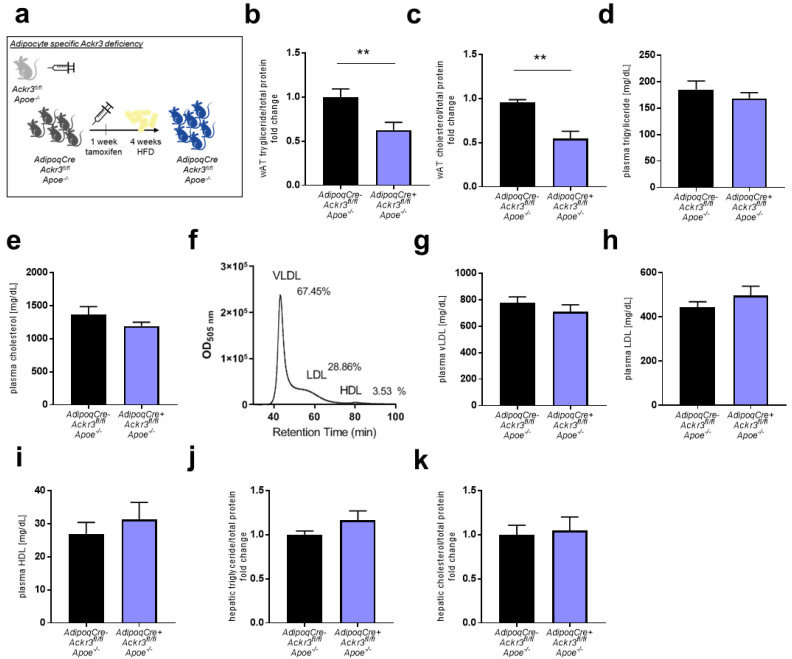
Atypical chemokine receptor 3 (ACKR3) deficiency reduces adipose tissue lipid levels. (**a**). Schematic representation of the experimental setup. (**b**,**c**). Total triglyceride (**b**, *n* = 12–14) and total cholesterol (**c**, *n* = 9–10) content in visceral white adipose tissue (wAT) samples normalized to total protein levels. (**d**,**e**). Total triglyceride (**d**) and total cholesterol (**e**) levels in the plasma of mice (*n* = 13–14). (**f**). Representative HPLC chromatogram of plasma lipoprotein fractions (very-low-density lipoprotein (VLDL), low-density lipoprotein (LDL), and high-density lipoprotein (HDL)). (**g**–**i**). Plasma levels of VLDL (**g**, *n* = 12–14), LDL (**h**, *n* = 12–14), and HDL (**i**, *n* = 12–14) levels. (**j**,**k**). Total triglyceride (**j**, *n* = 12–14) and total cholesterol (**k**, *n* = 12–14) levels in liver samples normalized to total protein levels. ** *p* < 0.01.

**Figure 2 biomedicines-09-00394-f002:**
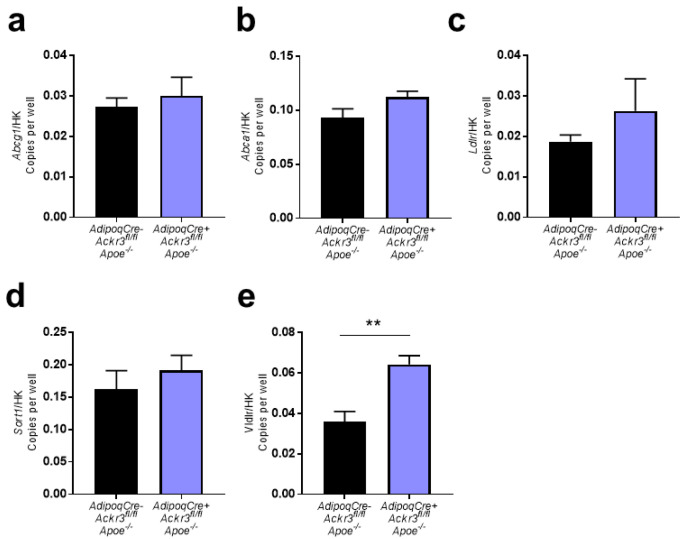
Expression of lipid receptors in wAT samples. a–e. Relative gene expression of *Abcg1* (**a**), *Abca1* (**b**), *Ldlr* (**c**), *Sort1* (**d**), and *Vldlr* (**e**) to housekeeping genes (*Gapdh* or *18S*) analyzed on Droplet Digital PCR (ddPCR) (all *n* = 4–7). ** *p* < 0.01.

**Figure 3 biomedicines-09-00394-f003:**
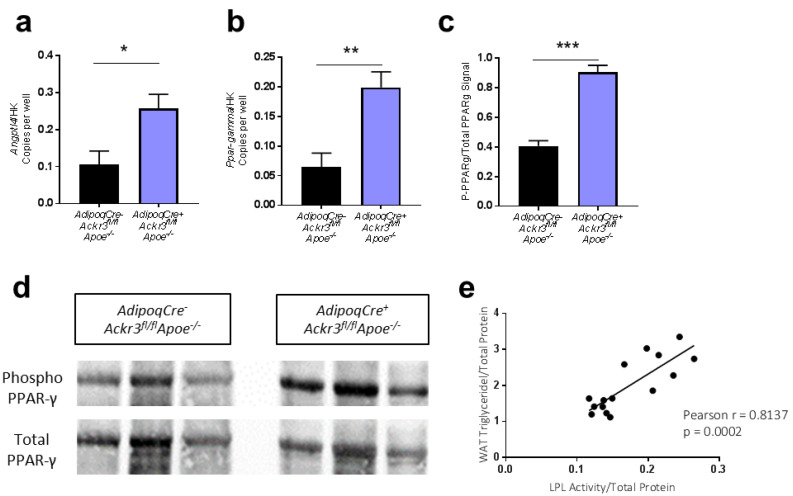
ACKR3 modulates lipid levels in wAT via ANGPTL4 and PPARγ. (**a**,**b**). Relative gene expression of *Angptl4* (**a**) and *Pparg* (**b**) to housekeeping genes (*Gapdh* or *18S*) analyzed on ddPCR (all *n* = 4–7). (**c**). Quantification of the western blot phospho-PPAR-γ signal normalized to total PPAR-γ signal on Image-J Software (*n* = 4). (**d**). Representative images of phospho-PPAR-γ and total PPAR-γ protein expression in wAT samples analyzed using western blot (Appendix A). (**e**). Pearson r correlation between wAT total triglyceride content and wAT LPL activity (r = 0.8137, *p* = 0.0002), (*n* = 15). * *p* < 0.05, ** *p* < 0.01, *** *p* < 0.001.

**Figure 4 biomedicines-09-00394-f004:**
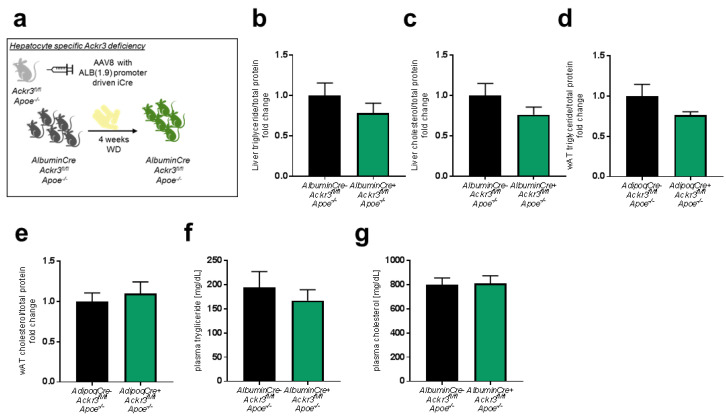
ACKR3 deficiency in hepatocytes does not influence hepatic lipid levels. (**a**). Schematic representation of the experimental setup. (**b**,**c**). Total triglyceride (**b**, *n* = 11–12) and total cholesterol (**c**, *n* = 11–12) levels in the liver. (**d**,**e**). Total triglyceride (**d**, *n* = 9–10) and total cholesterol (**e**, *n* = 9–10) content in wAT samples normalized to total protein levels. (**f**,**g**). Plasma levels of total triglyceride (**f**, *n* = 10–11) and total cholesterol (**g**, *n* = 10–11).

**Table 1 biomedicines-09-00394-t001:** Quantification of circulating immune cells, plasma cytokines, mouse, and wAT weights.

4-Week WD	AdipoqCre^−^Ackr3^fl/fl^Apoe^−/−^	AdipoqCre^+^Ackr3^fl/fl^Apoe^−/−^	*p*-Value
Leukocytes [×10^6^/mL]	2.4 ± 0.2	2.1 ± 0.1	0.3445
Neutrophils [×10^5^/mL]	6.5 ± 0.5	5.8 ± 0.5	0.3701
Classical Monocytes [×10^5^/mL]	1.9 ± 0.3	2.1 ± 0.2	0.5925
Non-classical Monocytes [×10^4^/mL]	7.7 ± 1.3	9.3 ± 1.7	0.5826
B cells [×10^5^/mL]	8.7 ± 1.1	7.2 ± 0.6	0.2358
T cells [×10^5^/mL]	3.1 ± 0.3	2.5 ± 0.2	0.1476
Plasma IL-6 [pg/mL]	6.0 ± 1.5	3.5 ± 1.0	0.3845
Plasma TNF-α [pg/mL]	11.6 ± 1.7	10.4 ± 0.6	0.9157
Plasma CXCL12 [pg/mL]	452.1 ± 50.2	386.3 ± 66.08	0.4360
Mouse weight [g]	24.5 ± 0.5	25.6 ± 0.7	0.2575
wAT weight [mg]	341.6 ± 62.71	384.6 ± 72.18	0.6633

## Data Availability

All data presented in this study are available upon reasonable request from the corresponding author.

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
