# Peer review of "Adipocyte-Specific ACKR3 Regulates Lipid Levels in Adipose Tissue"

_biomedicines, 2021, doi:10.3390/biomedicines9040394_

Round 1

Reviewer 1 Report

In the manuscript entitled “Adipocyte-specific ACKR3 regulates lipid levels in adipose tissue” by Selin Gencer et al., the authors demonstrated that ACKR3 regulates Ppar-gamma and Angptl4 expression in adipose tissue, thereby reducing the lipid level of adipose tissue. However, there is a lack of results and the conclusion does not seem to make sense in the manuscript. Thus, the manuscript should be improved. I would like to see additional experiments.

Major comments

  1. In figure 1, ACKR3 deficiency reduced adipose tissue lipid level. The author should show ACKR3 knockout of adipose tissues affects whether the body weight, the weight of adipose tissue, the resistance of insulin, and the levels of inflammatory cytokines.
  2. The authors need to explain the physiological significance of ACKR3 knockout of adipose tissue.
  3. In figure 3, PPAR-gamma was activated in the wAT of Ackr3 knockout mice. The authors should discuss how ACKR3 regulates the activation of PPAR-gamma.

Author Response

Reviewer 1

In the manuscript entitled “Adipocyte-specific ACKR3 regulates lipid levels in adipose tissue” by Selin Gencer et al., the authors demonstrated that ACKR3 regulates Ppar-gamma and Angptl4 expression in adipose tissue, thereby reducing the lipid level of adipose tissue. However, there is a lack of results and the conclusion does not seem to make sense in the manuscript. Thus, the manuscript should be improved. I would like to see additional experiments.

We would like to thank the reviewer for his/her evaluation of our manuscript. Based on the suggestion of extensive English editing, a native English speaker (British) has reviewed our manuscript in order to improve the quality of the English language and style of our paper. Furthermore, we made substantial adjustments to both the introduction as well as the conclusion sections to down-tune our conclusions a bit and especially to make them more context specific.

Major comments.

  1. In figure 1, ACKR3 deficiency reduced adipose tissue lipid level. The author should show ACKR3 knockout of adipose tissues affects whether the body weight, the weight of adipose tissue, the resistance of insulin, and the levels of inflammatory cytokines.

Thank you for this suggestion. We have now included the requested information regarding the body weight, adipose tissue weight and the levels of inflammatory cytokines in Table 1 which is now added to the manuscript. In order to provide further insight as to possible inflammatory status changes of the mice, we also included the quantities of circulating immune cells measured from the whole blood of mice via flow cytometry.

We thank the reviewer for raising the point of insulin resistance. Although we agree that this is an important and relevant parameter, we believe that this aspect is beyond the scope of the current manuscript. We hope that the current data presented in our manuscript will encourage further studies regarding ACKR3's potential roles in further relevant disease models, including insulin resistance and diabetic studies.

2. The authors need to explain the physiological significance of ACKR3 knockout of adipose tissue.

Although information regarding ACKR3's roles in adipocytes is very scarce in the literature, based on our results we included a discussion about the potential physiological significance of our findings (Lines 347-353).

3. In figure 3, PPAR-gamma was activated in the wAT of Ackr3 knockout mice. The authors should discuss how ACKR3 regulates the activation of PPAR-gamma.

Based on our novel findings and observations from previous studies, it can be suggested that there is a potential feedback mechanism between ACKR3 and PPAR-γ. We have elaborated on this mechanistic aspect in the discussion (Lines 318-329) as this will be an interesting focus point in future studies.

Reviewer 2 Report

This is an interesting study showing adipocyte specific regulatory effect of ACKR3 on lipid metabolism using the ApoE deficient mouse model. The study is well designed and generated some novel and potentially interesting aspects. However, it require some improvement to be suitable for publication.

1-Discussion: Please rewrite the first paragraph. The authors need to focus on their findings and use the literature to reflect and discuss what they found. Currently, they wrote another introduction at the beginning of the discussion (Line 247-264). Please start discussing at the third paragraph(Line 265), and use the earlier paragraphs to reflect on your findings.

2- Methods: Please be more succinate on obvious methods and give details on specific aspects (i.e. paragraph about Western Blot: it is too long and contains a lot of basic experimental steps, it would be suffice to state following  standard procedures except of the key steps. On the other hand, under HPLC, the authors mentioned separation of different lipoprotein fractions, they need to elaborate more, what are these fractions).

3-Formatting and style:  Figures 1&4: remove schematic from the main figure. State signification under all Figures (explain *,** levels under the figure, they were presented udder methods but you need to present under figures). Figure 3: Correlation between LPL activity and AT, This important finding need to be reflected on in the main result not only in the caption to the Figure.

4-Others: The paper is generally well-written, there is occasionally some repetition(Line 85-89: WD containing 21% fat and 0.15% to 0.2% cholesterol (Sniff diets). Also, supplementary data, I did not find any, probably due to technical issues.

Author Response

Reviewer 2

This is an interesting study showing adipocyte specific regulatory effect of ACKR3 on lipid metabolism using the ApoE deficient mouse model. The study is well designed and generated some novel and potentially interesting aspects. However, it require some improvement to be suitable for publication.

We would like to thank the reviewer for his/her evaluation of our manuscript and have implemented several improvements as described below.

1-Discussion: Please rewrite the first paragraph. The authors need to focus on their findings and use the literature to reflect and discuss what they found. Currently, they wrote another introduction at the beginning of the discussion (Line 247-264). Please start discussing at the third paragraph(Line 265), and use the earlier paragraphs to reflect on your findings.

As suggested by the reviewer, we have altered the arrangement of our discussion and tailored it as requested. Furthermore, we have elaborated the discussion of our results in light of the literature.

2- Methods: Please be more succinate on obvious methods and give details on specific aspects (i.e. paragraph about Western Blot: it is too long and contains a lot of basic experimental steps, it would be suffice to state following  standard procedures except of the key steps. On the other hand, under HPLC, the authors mentioned separation of different lipoprotein fractions, they need to elaborate more, what are these fractions).

In accordance to the suggestions above, we added further detail to the section of HPLC (Lines 130-137) and we shortened the Western blot section (Lines 164-189) in the methods. Meanwhile, further sections have been added in a condensed manner to the methods to described experiments that have been performed in the revision stage.

3-Formatting and style:  Figures 1&4: remove schematic from the main figure. State signification under all Figures (explain *,** levels under the figure, they were presented udder methods but you need to present under figures). Figure 3: Correlation between LPL activity and AT, This important finding need to be reflected on in the main result not only in the caption to the Figure.

As requested, we stated the signification under all figures in addition to the methods. Moreover, although this was mentioned briefly in the results section, we strengthened the emphasis on the correlation between LPL and AT lipid in the results section (Lines 261-266).

We thank the reviewer for suggesting the removal of  the schematics from the main figures, however we feel that it is a good overview of the experimental setup which makes it easier to convey our message to a general audience like the readers of Biomedicines, in addition to the color coding between adipocyte and hepatocyte specific knockouts.

4-Others: The paper is generally well-written, there is occasionally some repetition(Line 85-89: WD containing 21% fat and 0.15% to 0.2% cholesterol (Sniff diets). Also, supplementary data, I did not find any, probably due to technical issues.

We apologize for the inconvenience of not having the supplementary data which indeed seems to be a technical issue as we made sure to upload it during the submission process. The supplementary data showed the knockout confirmation and the full picture of the western blot images, therefore no additional significant data was involved in addition to the main figures. We had the paper reviewed in order to eliminate language errors and repetition as mentioned above.

Round 2

Reviewer 1 Report

The revised manuscript has been much improved and is in a nice condition now.